# Interactions among Genetic Background, Anesthetic Agent, and Oxygen Concentration Shape Blunt Traumatic Brain Injury Outcomes in *Drosophila melanogaster*

**DOI:** 10.3390/ijms21186926

**Published:** 2020-09-21

**Authors:** Amanda R. Scharenbrock, Hannah J. Schiffman, Zachariah P. G. Olufs, David A. Wassarman, Misha Perouansky

**Affiliations:** 1Department of Anesthesiology, University of Wisconsin-Madison, Madison, WI 53792, USA; ascharenbro2@wisc.edu (A.R.S.); hannah.j.schiffman@gmail.com (H.J.S.); olufs@wisc.edu (Z.P.G.O.); 2Department of Medical Genetics, University of Wisconsin-Madison, Madison, WI 53706, USA; dawassarman@wisc.edu

**Keywords:** anesthesia, *Drosophila melanogaster*, hyperoxia, isoflurane, pharmacogenetics, sevoflurane, secondary injury, TBI

## Abstract

Following traumatic brain injury (TBI), the time window during which secondary injuries develop provides a window for therapeutic interventions. During this time, many TBI victims undergo exposure to hyperoxia and anesthetics. We investigated the effects of genetic background on the interaction of oxygen and volatile general anesthetics with brain pathophysiology after closed-head TBI in the fruit fly *Drosophila melanogaster*. To test whether sevoflurane shares genetic risk factors for mortality with isoflurane and whether locomotion is affected similarly to mortality, we used a device that generates acceleration–deceleration forces to induce TBI in ten inbred fly lines. After TBI, we exposed flies to hyperoxia alone or in combination with isoflurane or sevoflurane and quantified mortality and locomotion 24 and 48 h after TBI. Modulation of TBI–induced mortality and locomotor impairment by hyperoxia with or without anesthetics varied among fly strains and among combinations of agents. Resistance to increased mortality from hyperoxic isoflurane predicted resistance to increased mortality from hyperoxic sevoflurane but did not predict the degree of locomotion impairment under any condition. These findings are important because they demonstrate that, in the context of TBI, genetic background determines the latent toxic potentials of oxygen and anesthetics.

## 1. Introduction

Traumatic brain injury (TBI) is frequently the determining factor for the most debilitating outcomes from polytrauma [1]. The consequences of TBI result from both primary and secondary injuries to the brain. Secondary cellular injuries develop over a variable period of time after primary physical injuries. This period is crucial because it offers a window of opportunity for pharmacologic interventions to improve (or unintentionally aggravate) short- and long-term outcomes [2].

To date, no pharmacologic therapy aimed at the secondary injury phase of TBI has consistently improved clinical outcomes. One possible reason for the lack of success may be genetic heterogeneity among individuals, resulting in variable effectiveness of targets for pharmacologic intervention [3]. If so, genetically accessible and affordable models of TBI are bound to make an important contribution to personalized care of TBI patients [4].

We have studied effects of anesthetics on mortality within 24 h following closed-head TBI in a *Drosophila melanogaster* (fruit fly) model [5,6,7]. The fly TBI model uses a spring-based instrument to inflict blunt force polytrauma to adult flies. Several lines of evidence indicate that mortality and impaired locomotion in this model are due to primary injuries to the brain. We used a serial anesthesia array to expose injured and uninjured flies to precise doses of the volatile general anesthetics (VGAs) isoflurane (ISO) and sevoflurane (SEVO) and the desired concentration of oxygen (O_2_) [8]. We found that exposure to ISO or ISO and 100% O_2_ (ISO/O_2_) increases mortality after TBI in the *w^1118^* laboratory fly strain [8,9]. We also found that genetic background affects the extent to which exposure to ISO/O_2_ increases mortality following TBI. Examination of 141 inbred fly lines from the *Drosophila* Genetic Reference Panel (DGRP) collection revealed that exposure to ISO/O_2_ following TBI enhances mortality to varying degrees, from no effect to an increase of 34 percentage points [9]. Deleterious effects may be due to anesthetic-induced neurotoxicity (AiN), which has been associated with adverse neuropsychological consequences in higher animals [10].

Building upon these findings, we investigated effects of exposure to O_2_ or SEVO/O_2_ on mortality following TBI. We also examined effects of exposure to O_2_, ISO/O_2_, or SEVO/O_2_ on impairment of locomotion following TBI. Mortality is an unambiguous marker of AiN, and impairment of locomotion is a non-lethal phenotype of AiN relevant for survivors of TBI. To focus the study, we selected ten lines from the DGRP collection, the five most resilient and the five most susceptible to increased mortality from exposure to ISO/O_2_ [9]. We found that effects of exposures on mortality and locomotion following TBI were not correlated, that exposures had different effects on mortality and locomotion between the two groups of DGRP lines, and that O_2_, ISO/O_2_, and SEVO/O_2_ had line-specific effects on mortality and locomotion. The data suggest that the fruit fly is a useful model to explore the pharmacogenetic complexity of TBI outcomes in higher animals.

## 2. Results

We investigated the effects that exposure to anesthetics (ISO and SEVO) and hyperoxia (100% O_2_) have on TBI outcomes (mortality and locomotor impairment) in ten fly lines (RAL lines) from the DGRP collection that have different genetic backgrounds [11]. Figure 1 and 5 provide overviews of the mortality and locomotor impairment (impaired climbing) data, respectively, at 24 h after TBI. Figure 2, Figure 3 and Figure 4 and 6–8 analyzed the data from Figure 1 and 5 in distinct ways that highlight particular aspects of the response to anesthetic/hyperoxia exposure following TBI. In Figure 2 and 6, the ten RAL lines were analyzed as two groups that were classified based on our prior analysis of the effect of exposure to ISO/hyperoxia on mortality following TBI [9]. In contrast, Figure 3 and 7 analyzed individual RAL lines, and Figure 4 and 8 analyzed the relationship between mortality and locomotor impairment for individual RAL lines. Figure A1, Figure A2 and Figure A3 and Figure A14, Figure A15 and Figure A16 provide analogous data at 48 h after TBI. In all experiments, TBI was inflicted in room air (normoxia) using a standard protocol consisting of four strikes from the High-Impact Trauma (HIT) device with 5 min intervals between strikes. After TBI, flies received either 100% oxygen (O_2_) or equipotent concentrations of ISO or SEVO in 100% O_2_ (O_2_, ISO/O_2_, and SEVO/O_2_, respectively) for 1 h, while control flies remained in normoxia (see Material and Methods for more details). 

### 2.1. Genetic Background Modulates the Effect of Anesthetic Exposure on Mortality Following TBI

Based on results from our prior screen of 141 RAL lines from the DGRP collection, we selected ten RAL lines to analyze in more detail. Five lines had the largest excess mortality from ISO/O_2_ toxicity, that is the highest percentage point increase in mortality between flies that were and those that were not exposed to ISO/O_2_ following TBI. We refer to these lines as the high ∆ group. The other five lines had the lowest excess mortality (percentage point increase in mortality), and we refer to these lines as the low ∆ group. Among the ten lines, the ∆ ranged from 0% (RAL332) to 34% (RAL859). 

Figure 1 shows the mortality of the ten RAL lines 24 h after TBI alone or after TBI followed by a 1 h exposure to O_2_, ISO/O_2_, or SEVO/O_2_. RAL lines are arranged in Figure 1 from left to right with increasing mortality from TBI alone. We report mortality as the mortality index at 24 h, which is the percent mortality of injured flies normalized to the percent mortality of uninjured flies 24 h following TBI. As previously reported, lines in the high ∆ group had lower mortality after TBI alone than lines in the low ∆ group. Furthermore, exposure to ISO/O_2_ following TBI increased mortality of lines in the high ∆ group to a level that was similar to lines in the low ∆ group. In contrast, exposure to O_2_ or SEVO/O_2_ did not have the same group-wide effects on the high ∆ group as exposure to ISO/O_2_. For example, exposure of lines in the high ∆ group to SEVO/O_2_ did not increase mortality for some lines (RAL761, RAL491, and RAL859), and exposure to O_2_ did not increase mortality to the same extent as exposure to ISO/O_2_ for some lines (RAL491 and RAL859) but did for the other lines (RAL382, RAL761, and RAL373), whereas exposure of lines in the low ∆ group to O_2_ or SEVO/O_2_ largely mimicked the lack of an effect of exposure to ISO/O_2_ on mortality, the exception being increased mortality of RAL83 flies exposed to O_2_. Nearly identical genotype-specific effects of exposure to O_2_, ISO/O_2_, and SEVO/O_2_ on mortality at 24 h following TBI were observed at 48 h following TBI (Appendix A
Figure A1). Therefore, analyses of flies with different genetic backgrounds revealed drug-specific pharmacogenetic contributions to mortality at 24 and 48 h following TBI.

### 2.2. Effects of ISO/O_2_ Exposure on Mortality Following TBI Stratifies Effects of O_2_ and SEVO/O_2_ Exposure on Mortality

In Figure 2, we analyzed the MI_24_ data from Figure 1 based on the classification of RAL lines into high ∆ and low ∆ groups. This analysis shows that the high ∆ group had a significantly lower MI_24_ than the low ∆ group following TBI alone (16.0 ± 2.4% and 37.8 ± 8.7%, respectively, *p* = 0.0005). Moreover, using the MI_24_ as a readout, the high ∆ and low ∆ groups had significantly different responses to exposure to O_2_, ISO/O_2_, and SEVO/O_2_ (*p* = 0.0248 for treatment effect, two-way analysis of variance (ANOVA)). Within the high ∆ group, exposure to ISO/O_2_ led to significantly higher mortality than exposure to SEVO/O_2_ (*p* = 0.0059), while exposure to O_2_ or ISO/O_2_ produced similar mortality (*p* = 0.1386). In contrast, within the low ∆ group, there were no significant differences in mortality among the exposures. Analogous analyses of the MI_24_ at 48 h post-TBI led to the same conclusions (Appendix A
Figure A2). We conclude that resilience to TBI incurs heightened susceptibility of survivors to subsequent stress from exposure to O_2_ and anesthetics.

We also observed that the MI_24_ of lines in the high ∆ group were tightly clustered after TBI (SD = 2.4) but more variable among lines in the low ∆ group (SD = 8.7). Exposure to O_2_ substantially increased variability of the MI_24_ in the low ∆ but not in the high ∆ group (SD = 20.2 and 5.7, respectively), indicating that the averaged group response conceals a high degree of variability in the response to hyperoxia after TBI between individual RAL lines in the low ∆ group.

### 2.3. Variation in the MI_24_ of Individual Lines Following O_2_, ISO/O_2_, and SEVO/O_2_ Exposure after TBI Is More Common in the High ∆ Group than the Low ∆ Group

In Figure 3, we analyzed the MI_24_ data from Figure 1 with respect to individual RAL lines within the high ∆ and low ∆ groups. For all lines within the high ∆ group, exposure to O_2_, ISO/O2, or SEVO/O_2_ after TBI had a significant effect on the MI_24_ (*p* < 0.0001 for each line, one-way ANOVA) (Figure 3a). In contrast, for lines within the low ∆ group, exposure after TBI only affected RAL83, for which exposure to O_2_ significantly increased the MI_24_ (*p* = 0.0018, one-way ANOVA) (Figure 3b). Analysis of the MI_24_ at 48 h following TBI and exposure to O_2_ or SEVO/O_2_ led to the same findings (Appendix A
Figure A3). We conclude that the high ∆ and low ∆ classification captures pharmacogenetic properties shaping the mortality phenotype in response to O2 and VGAs.

### 2.4. Locomotion of Uninjured Flies Has a Weak Negative Correlation with Mortality of Flies Following TBI

To further assess the effects of anesthetics and hyperoxia on TBI outcomes in different genetic backgrounds, we used a climbing assay to monitor the locomotion of flies. In brief, the climbing assay determined the percent of flies that were unable to climb 2.5 cm in 10 s after being gently tapped to the bottom of the vial. For each of the ten RAL lines, we assayed climbing of uninjured flies to establish the baseline level of locomotion. Figure 4 revealed a weak correlation (R^2^ = 0.314) between the percent impaired climbing of uninjured flies and the MI_24_ of flies following TBI. These data indicate that fly lines with a low percent impaired climbing (i.e., high locomotion activity) tend to be more susceptible to mortality following TBI.

### 2.5. Genetic Background Modulates the Effect of Anesthetic Exposure on Locomotion Following TBI

To further investigate the effects that exposure to anesthetics and hyperoxia have on TBI outcomes in different genetic backgrounds, we measured percent impaired climbing for the ten RAL lines after TBI alone as well as after TBI followed by exposure to O_2_, ISO/O_2_, or SEVO/O_2_. Flies were assayed for climbing activity at 24 h and 48 h after the treatment condition. We determined the percent of flies that were able to climb to each of four 2.5 cm quadrants in a fly vial within 10 s after being gently tapped to the bottom of the vial. These data are presented in Appendix A
Figure A4, Figure A5, Figure A6, Figure A7, Figure A8, Figure A9, Figure A10, Figure A11, Figure A12 and Figure A13. Figure 5, Figure 6, Figure 7 and Figure 8 focus on analyses of flies with impaired climbing, that is the fraction of flies that were unable to climb beyond the lowest quadrant. In Figure 5, the RAL lines are arranged in the same order from left to right as in Figure 1. For lines such as RAL382, TBI and exposure to agents had no effect on climbing ability, whereas for lines such as RAL707, TBI and exposure to any agent impaired climbing ability (i.e., a larger percent of flies remained in the bottom quadrant). There were also lines such as RAL373 whose climbing ability was differentially affected by TBI and exposure to different agents. On repeat testing at 48 h after TBI, climbing impairment changed in comparison to that recorded at 24 h in three lines (Appendix A
Figure A14). Climbing performance recovered somewhat after ISO/O_2_ exposure in RAL373 and RAL374 (*p* = 0.0212 and <0.0001, respectively) and further deteriorated in RAL859 after O_2_ and ISO/O_2_ (*p* = 0.0314 and 0.0395, respectively). These data indicate that genetic background affects the extent to which exposure to O_2_, ISO/O_2_, or SEVO/O_2_ alters locomotion following TBI.

### 2.6. ISO/O_2_ and SEVO/O_2_ Exposure Impairs Climbing of the Low ∆ Group Following TBI 

For both uninjured and injured flies, the high ∆ and low ∆ groups had similar responses to exposure to O_2_, ISO/O_2_, and SEVO/O_2_ (*p* = 0.5938 and *p* = 0.6083 for interaction and group, two-way ANOVA) (Figure 6a,b). However, there was more variability in climbing ability among RAL lines in the high ∆ group than in the low ∆ group. Moreover, for injured flies, exposure to ISO/O_2_ or SEVO/O_2_ significantly impaired the climbing ability of the low ∆ group (*p* = 0.008, one-way ANOVA) (Figure 6b). Neither the high ∆ nor the low ∆ group had significant differences at 48 h post-TBI (Appendix A
Figure A15). Therefore, cellular processes that drive the classification of RAL lines into high ∆ and low ∆ groups based on effects of ISO/O_2_ exposure on mortality following TBI also appear to differentiate the groups with respect to locomotion for both uninjured and injured flies exposed to O_2_, ISO/O_2_, or SEVO/O_2_.

### 2.7. Percent Impaired Climbing of Individual Lines within the High ∆ Group but Not Low ∆ Groups Varies Following O_2_, ISO/O_2_, or SEVO/O_2_ Exposure after TBI

In contrast to the lack of effect on the group level, examination of individual fly lines revealed substantial variability in the responses to the three treatments after TBI. Neither TBI nor treatments had a significant impact on locomotor performance in two lines of the high ∆ group, (RAL382 and RAL491) (Figure 7a). O_2_ treatment improved climbing in one line (RAL373), but ISO/O_2_ significantly worsened it (*p* = 0.0057 and 0.0012, unpaired two-tailed *t*-test). By contrast, climbing in RAL859 deteriorated substantially from exposure to O_2_ and was partially rescued by ISO/O_2_ and SEVO/O_2_. (*p* < 0.0001, one-way ANOVA). At 48 h, O_2_ treatment no longer affected the climbing of lines RAL373 and RAL859 (Appendix A
Figure A16A). Locomotion in all lines of the low ∆ group deteriorated to a similar degree after TBI and showed no significant changes with any exposure (Figure 7b). Similar findings were observed at 48 h (Appendix A
Figure A16B). We conclude that genetic background influences the locomotion phenotype of AiN after TBI and that the nature of the effect is not predicted by the mortality phenotype. 

### 2.8. The Weak Correlation between the MI_24_ and Locomotion Impairment Is Lost after Interventions

While the MI_24_ modestly correlated with baseline climbing activity (Figure 4), this correlation was absent when exposure to any of the agents followed TBI (Figure 8).

## 3. Discussion

Exposure to general anesthetics may be deleterious for the developing and the degenerating brain [10], and hyperoxia is associated with adverse outcomes after brain injury in some (but not all) clinical studies [12,13,14,15]. A pharmacogenetic strategy to determine whether genetic background affects the risk of toxicity from these commonly employed therapeutic strategies seems warranted. Our results show that genetic background influences the effects of exposure to VGAs and hyperoxia on two outcomes after TBI.

### 3.1. Resilience to TBI-Induced Mortality Correlates Inversely with Susceptibility to AiN

We found, serendipitously, that susceptibility to AiN after TBI correlated with resilience to mortality from TBI per se (Figure 1 and Figure 2). Fly lines with high mortality from TBI suffered little excess mortality from exposure to anesthetics and hyperoxia, and the opposite was true for lines with low mortality from TBI. These observations suggest that exposure to anesthetics and O_2_ after TBI may follow a two-hit model in which a substantial fraction of survivors in a relatively resilient population finds itself at high risk from additional stressors, while in a population severely impacted by the first hit, surviving individuals are relatively tolerant to a second stressor. This correlation was phenotype specific, as it did not translate to impairment of locomotion.

### 3.2. VGA-Toxicity Is Distinguishable from O_2_-Toxicity

VGAs are (almost) invariably administered in supranormal oxygen concentrations, but hyperoxia is frequently administered without concomitant use of VGAs. Separating the noxious effects of VGAs from those of hyperoxia is therefore important for risk assessment. We found that in select genetic backgrounds, O_2_ and ISO toxicity were clearly distinguishable, with the toxicity of ISO/O_2_ being lower than that of O_2_. Interestingly, this separation was observable in different genetic backgrounds for the two phenotypes. RAL382 and RAL83 showed markedly increased mortality from exposure to hyperoxia (Figure 3) with mitigating effects from co-administration of either ISO or SEVO, indicating that in these genetic backgrounds O_2_ is a strong toxin in the context of TBI. However, neither line showed an analogous effect on locomotion. By contrast, in RAL859, ISO/O_2_ resulted in higher mortality than O_2_, and SEVO/O_2_ had no effect on mortality. Locomotion, by contrast, was more impaired by O_2_ than by either ISO/O_2_ or SEVO/O_2_. Finally, RAL373 had the worst locomotion outcome from exposure to ISO/O_2_ while O_2_ showed no adverse effect. These results indicate that the two phenotypes are different manifestations of toxicity and may be grounded in different molecular processes.

Both phenotypes, however, share common ground in the effect of SEVO/O_2_. In both high ∆ and low ∆ groups, both increased mortality and impaired locomotion from SEVO/O_2_ were either lower or at least not higher than that from ISO/O_2_. We conclude that resilience to AiN from ISO/O_2_ predicts resilience to SEVO/O_2_ with the caveat that the RAL lines were selected based on excess mortality from ISO/O_2_. The DGRP collection may contain lines with different susceptibility profiles.

These results demonstrate for the first time that genetic background determines the pharmacodynamic interactions of two agents, ISO and SEVO, used interchangeably in clinical practice with O_2_.

### 3.3. Are Mortality and Impaired Locomotion Mechanistically Unrelated Phenotypes?

We found that the effect of O_2_ and the VGAs on mortality after TBI does not predict their effect on locomotion. Our experiments did not address the mechanisms underlying this dissociation but illustrate the complex relationships between different outcomes after TBI. It is noteworthy that mortality and locomotion profiles switch between the high and the low ∆ groups (compare Figure 3 and Figure 7). The low ∆ group has high variability for mortality among the lines, while the mortality within each line is largely unaffected by exposures. In contrast, for locomotion, it is the high ∆ group that has high variability among the lines, while locomotion within each line is insensitive to the different exposures. In general, the data in Figure 3 and Figure 7 show that anesthetics/hyperoxia either increase mortality or impair climbing but not do both in a single RAL line.

Attesting to the importance of pharmacogenetic determinants, neither TBI nor the exposures affected locomotion performance in RAL382. By contrast, in RAL83, locomotion was improved by hyperoxia after TBI but was deteriorated by ISO/O_2_. We interpret these results as confirming our conclusion gained from analyzing mortality in that ISO and O_2_ are independently noxious but, as the effects on mortality vs. locomotion diverge between fly lines, these phenotypes of TBI may reflect mechanistically independent manifestations of AiN.

### 3.4. The Modifying Effects of ISO/O_2_ on Mortality and Locomotion Following TBI in Different Genetic Backgrounds Does Not Predict the Modifying Effects of O_2_ or SEVO/O_2_

High excess mortality from ISO/O_2_ toxicity did not always predict the response to O_2_ and SEVO/O_2_ on mortality (Figure 3a). However, resilience to ISO/O_2_ toxicity correlated with resilience to SEVO/O_2_ (Figure 3b). These data indicate that either the molecular targets mediating toxicity of ISO and SEVO do not completely overlap or that ISO is more efficient at causing AiN at equi-behavioral doses.

An interesting exception was RAL83. While this line was highly sensitive to O_2_, addition of either VGA reduced mortality (Figure 3b), suggesting a complex interplay between O_2_, VGAs, and genetic background.

There was no correlation between mortality and climbing impairment (Figure 8). Interestingly, O_2_ after TBI resulted in opposing effects on climbing in two lines (RAL373 improved while RAL859 deteriorated) in the high ∆ group. 

### 3.5. Climbing Activity as Predictor of Resilience to TBI

We found that the high ∆ group showed relatively little climbing activity prior to TBI. This group also experienced lower mortality from TBI (Figure 4), possibly suggesting a link between baseline energy metabolism and resilience to TBI. The correlation between climbing activity in uninjured flies and the MI_24_ after TBI (Figure 4) was lost in injured animals (Figure 8a) and all exposures.

### 3.6. Genetic Background Confers Discrete Sensitivity/Resilience to TBI, VGAs, and O_2_

Our previous genome-wide association study (GWAS) analysis of 141 DGRP lines challenged with TBI and ISO/O_2_ identified five SNPs in three biologically plausible genes (*Prip, Drip* and *Gyc88E*) that were associated with variability in mortality at a *p*-value threshold *<*10^−7^ [9]. *Prip* and *Drip* are orthologous to mammalian water-permeable channels (aquaporins), and Gyc88E is orthologous to the oxygen sensor *GUCYB1*.

Data presented here also show that O_2_ is toxic in young animals with certain genetic backgrounds, suggesting a link to polymorphisms in *Gyc88E*. Previously, we found that in the standard *w^1118^* laboratory fly strain, O_2_ toxicity after TBI was detectable only in old flies [9]. Notably, however, while SEVO/O_2_ had no excess mortality in young flies, it was associated with a higher excess mortality than ISO/O_2_ in old flies. We interpret these findings as indicating that genetic background plays an important role in determining the susceptibility of the brain to the stresses of VGA/O_2_. The degree to which aging shapes this interaction in various genetic backgrounds remains unknown.

### 3.7. Summary

*D. melanogaster* is a recognized tool in neurotoxicologic research [16]. We have instrumentalized the DGRP collection to facilitate a pharmacogenetic analysis of the interaction between anesthesia and the injured brain. Our experiments provide a basis for future preclinical experiments aimed at the development of personalized care of TBI patients.

## 4. Materials and Methods 

The manuscript adheres to the applicable ARRIVE (Animal Research: Reporting of In Vivo Experiments) reporting guidelines (preclinical animal research). Approval from the Institutional Animal Care and Use Committee has been waived.

### 4.1. TBI

We inflicted TBI using a HIT device in room air [7]. All flies were maintained on molasses food at 25 °C. On the day of the experiment, flies were rapidly transferred into empty vials. The vials were subjected to four strikes from the HIT device with the spring deflected to 90 degrees and 5 min between strikes. After injury and exposures, flies were transferred to vials with molasses food and incubated at 25 °C. The mortality index was determined at 24 h from the time of TBI (MI_24_). Because the MI_24_ does not differ between male and female flies, we performed all experiments using mixed-sex groups [6]. 

### 4.2. Exposure to VGAs and O_2_

We used a custom-built Serial Anesthesia Array to simultaneously expose up to eight samples of at least 20 flies each to precise doses of VGAs and oxygen [8,17]. VGAs were administered through the Array using a Datex-Ohmeda Aestiva/5 anesthesia machine equipped with commercial agent-specific vaporizers (Datex-Ohmeda Inc., Madison, WI, USA). Compressed gas cylinders (Airgas USA, LLC, Radnor, PA, USA) containing 100% oxygen (O_2_), 100% nitrogen (N_2_) or air (21% O_2_/79% N_2_) provided carrier gas of the desired composition. Anesthetic exposures consisted of either 2% ISO or 3.5% SEVO for 1 h (i.e., 2%h and 3.5%h, respectively). The resulting anesthetic doses of 2%h and 3.5%h for ISO and SEVO, respectively, are behaviorally equivalent and do not affect median and maximum lifespans [8]. In experiments under hyperoxic conditions, 100% O_2_ was used as the carrier gas to administer 2% ISO or 3.5% SEVO. All flies resumed movement in less than 1 h after discontinuing ISO or SEVO, indicating that the doses were safe. After an experiment, flies were returned to food-containing vials in temperature-controlled humidified incubators. Mortality and locomotion were scored at 24 and 48 h post TBI. All experiments were conducted under normobaric conditions. 

### 4.3. Genetic Background

We tested 1–8 day old flies. The tested lines were selected from 141 DGRP lines previously tested [9] with the standard four-strike injury protocol, with post-exposure to 1 h of ISO in hyperoxia. Control flies were kept at room air. No mortality was recorded in any lines when the flies were not challenged with TBI or only exposed to anesthesia and hyperoxia. Among flies challenged with TBI and TBI followed by ISO/O_2_, RAL 332, 818, 707, 83, and 374 had the lowest excess mortality (∆). RAL 859, 373, 382, 491, and 761 had the highest ∆ from post-TBI exposure to ISO/O_2_, which we interpret as a sign of AiN. Each experiment included two vials of 20 flies each (considered a single biological replicate). Every experiment consisted of at least three biological replicates. The flies experienced only a single exposure and were discarded after recording the 48 h results.

### 4.4. High and Low ∆ Groups

Group assignment into high ∆ and low ∆ was based on the previously determined excess mortality after TBI when TBI was followed by a 1 h exposure to ISO/O_2_ [9]. In this project, the difference in mortality between anesthetized and unanesthetized populations was determined 24 and 48 h after TBI.

### 4.5. Mortality

The mortality index (MI_24_ and MI_48_) quantifies the fraction of flies found dead at 24 and 48 h after TBI. It subtracts out deaths due to natural attrition, which were virtually zero in all RAL lines for this age group. Excess mortality (∆) caused by exposure for 1 h to 100% O_2_, 2% ISO/O_2_ or 3.5% SEVO/O_2_ after TBI represents the average percentage point increase in mortality for flies with TBI followed by exposure to the agents compared to the percent mortality for flies with TBI only.

### 4.6. Impairment of Locomotion

We used a modification of the “rapid iterative negative geotaxis” (RING) [8] as a versatile and reproducible test of integrated neural activity. Gait speed correlates with impending cognitive impairment in humans [18], while in flies locomotor activity integrates multiple modalities indicative of health [19]. To assess the degree to which exposure to VGAs after TBI modulates long-term neural function, we took advantage of the innate escape response termed ‘negative geotaxis’. This response is elicited by mechanical stimulation: After being tapped to the bottom, flies climb up the wall of a container. RING ‘bandwidth’ can be adjusted (e.g., for populations with *a priori* impairment of motor function) and can easily be quantified to provide accurate measures of locomotion to quantify the influence of factors affecting integrated neural activity (e.g., age, mutations or drugs) [8,20]. For each replicate, four groups of 20 flies of the same age from a single RAL line were used. The groups were (i) naïve, (ii) TBI only, (iii) one of the three possible exposures (O_2_, ISO/O_2_, and SEVO/O_2_) only, and (iv) TBI followed by one of the three exposures. Flies were placed in a container subdivided into four zones along its height and tapped down. At 10 s, a photo of the container was taken and the number of flies in each zone counted. Climbing was examined at 24 h and 48 h after TBI. We used the fraction of live flies unable to leave the lowest quadrant as our measure of impairment (Percent Impaired Climbing).

### 4.7. Statistical Analysis

We used Prism 8.0 (GraphPad Software, Inc., La Jolla, CA, USA) for graphing. Box plots used to describe mortality data include the second and third quartiles of data (Q2 and Q3). The median (horizontal bar) is indicated, and whiskers extend to the minimum and maximum data points. Locomotor impairment refers to the comparison between TBI and exposures. We report numerical results as mean ± SD. We used parametric descriptive statistics because mortality rates after TBI for male and female flies were normally distributed. Statistical significance between two data points (control and experimental condition) was tested using an unpaired equal-variance two-tail Student’s *t*-test. We used one-way ANOVA for multiple comparisons within one genotype and two-way ANOVA for comparisons between genotypes and exposures.

## Figures and Tables

**Figure 1 ijms-21-06926-f001:**
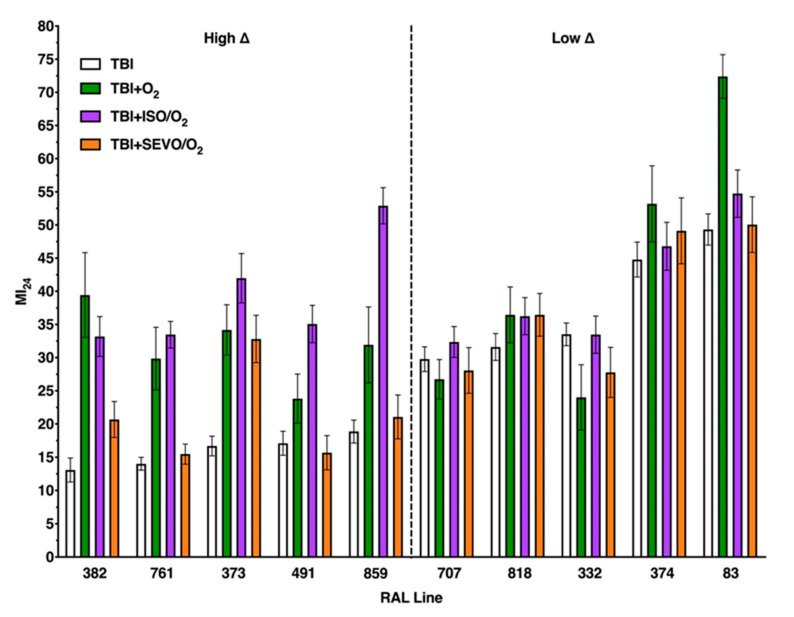
Genetic background alters the effect of exposure to O_2_, ISO/O_2_, or sevoflurane (SEVO)/O_2_ on mortality 24 h after traumatic brain injury (TBI). Ten fly lines (RAL lines) from the DGRP collection, indicated by RAL line number, are arranged from left to right by increasing MI_24_ from TBI. Exposure to ISO/O_2_ increased the MI_24_ of high ∆ lines (left) but not low ∆ lines (right). The ∆ mortality in percentage points between TBI alone and TBI with exposure ranged from 0 (RAL332 33.5 ± 11.1% vs. 33.5 ± 11.9%) to 34 (RAL859 18.9 ± 11.2% vs. 52.9 ± 11.6%) for ISO/O_2_; from −9.5 (RAL332 33.5 ± 11.1% vs. 24.0 ± 13.9%) to 26.3 (RAL382 13.1 ± 11.6% vs. 39.4 ± 18.1) for O_2_; and from −5.7 (RAL332 33.5 ± 11.1% vs. 27.7 ± 15.1%) to 16.1 (RAL 373 16.7 ± 9.8% vs. 32.8 ± 14.3%) for SEVO/O_2_. Bars represent the average MI_24_ and standard deviation of at least three independent samples of 20 mixed sex flies each at 1–8 days old.

**Figure 2 ijms-21-06926-f002:**
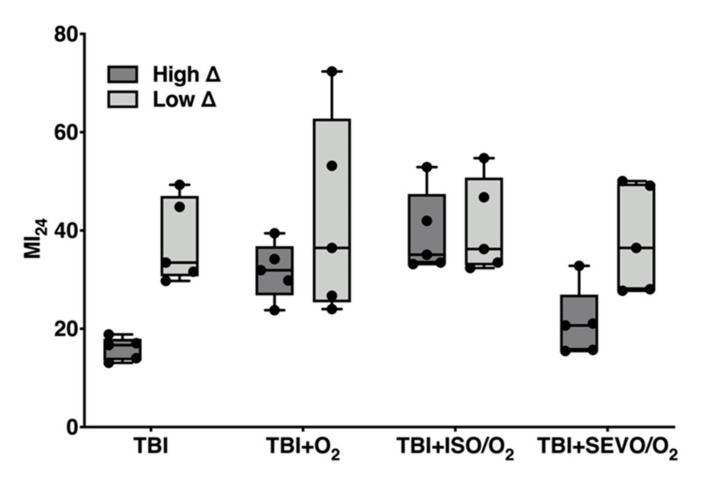
High ∆ and low ∆ groups respond differently to O_2_, ISO/O_2_, and SEVO/O_2_ exposure following TBI. Box plots of the MI_24_ for the five RAL lines in the high ∆ and low ∆ groups under the indicated treatment conditions. Horizontal bars indicate the median and whiskers extend to the minimum and maximum data points.

**Figure 3 ijms-21-06926-f003:**
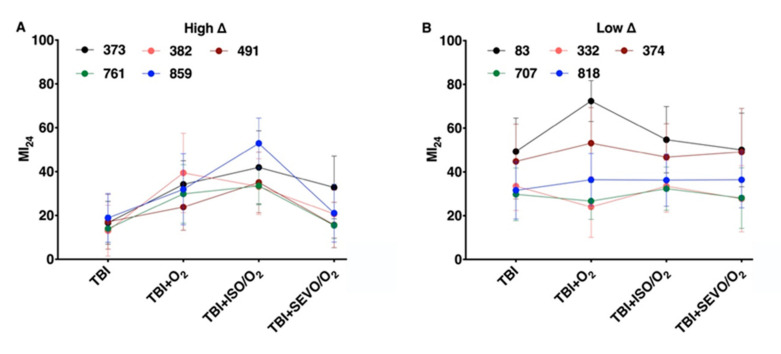
The response of a fly line to ISO/O_2_ exposure following TBI does not always predict its response to O_2_ or SEVO/O_2_ exposure. Colored dots and error bars indicate the average and standard deviation, respectively, of the MI_24_ for three independent samples of individual RAL lines under the indicated treatment conditions. (**A**) High ∆ lines and (**B**) low ∆ lines.

**Figure 4 ijms-21-06926-f004:**
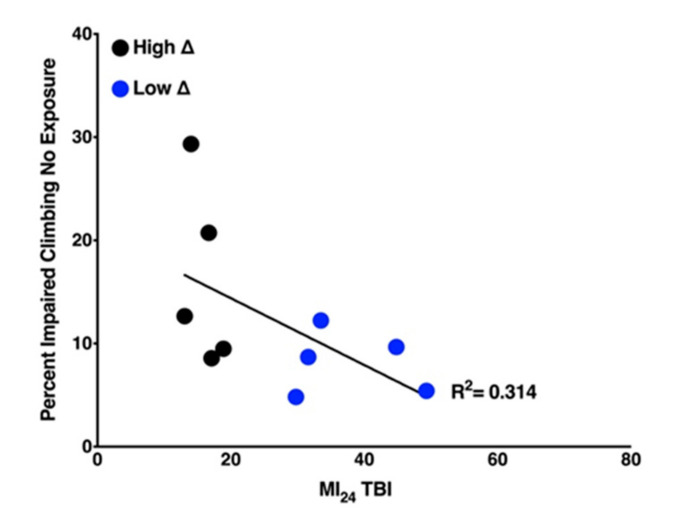
Mortality correlates negatively with climbing activity of uninjured flies. Percent impaired climbing of uninjured RAL lines is plotted against the MI_24_ of injured RAL lines from the high ∆ group (black dots) and low ∆ group (blue dots).

**Figure 5 ijms-21-06926-f005:**
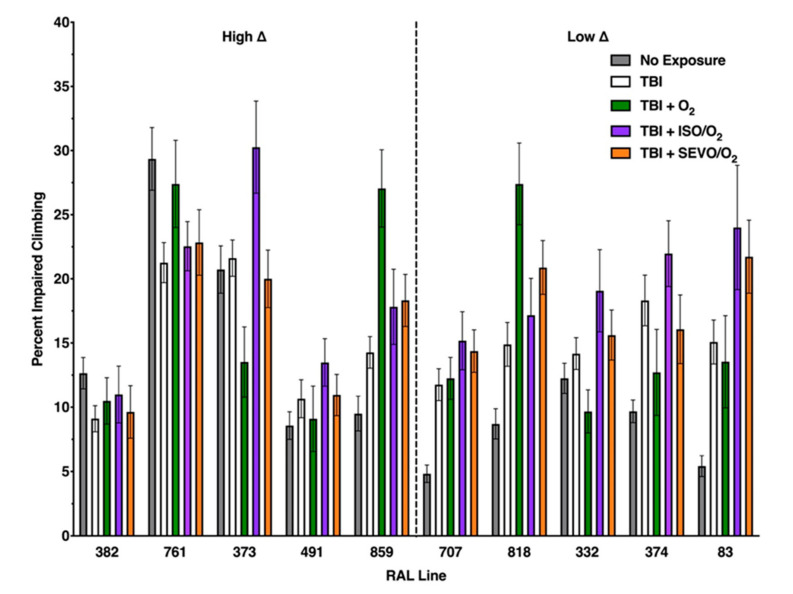
Genetic background alters the effect of exposure to O_2_, ISO/O_2_, or SEVO/O_2_ on percent impaired climbing 24 h after TBI. Percent impaired climbing was determined for the ten RAL lines that were analyzed in Figure 1, Figure 2, Figure 3 and Figure 4 and displayed in the same order as Figure 1 in high ∆ and low ∆ groups. Percent impaired climbing was determined for uninjured and unexposed flies (No Exposure) as well as injured flies and the same exposure conditions as Figure 1. Higher bars indicate more impairment.

**Figure 6 ijms-21-06926-f006:**
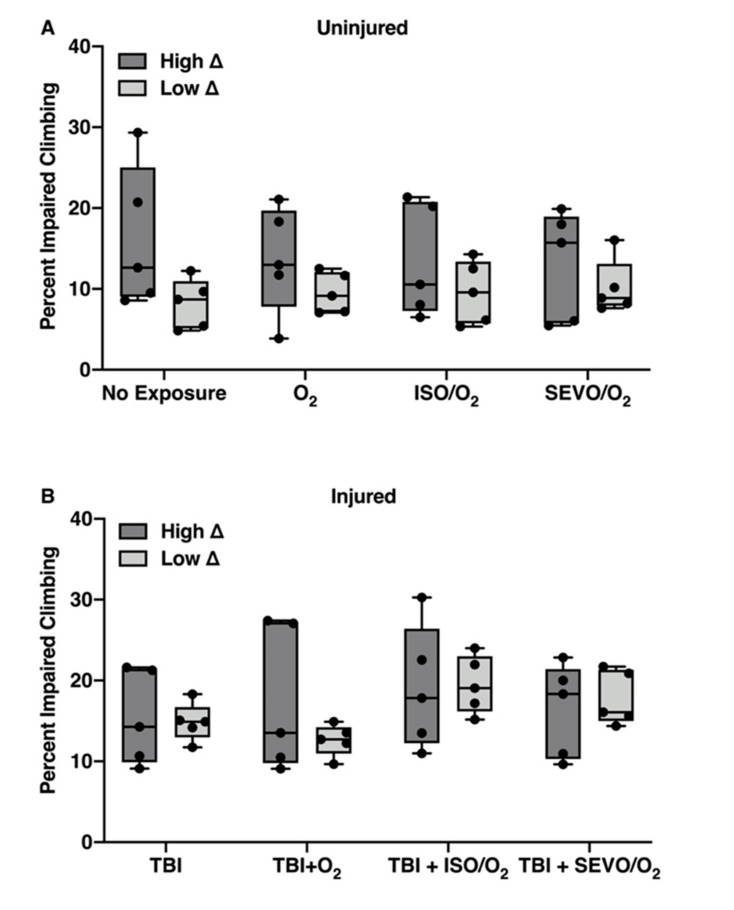
Classification into high ∆ and low ∆ groups predicts effects on locomotion. (**A**) Box plots of percent impaired climbing for uninjured flies from the five RAL lines in the high ∆ and low ∆ groups under the indicated treatment conditions. Horizontal bars indicate the median, and whiskers extend to the minimum and maximum data points. (**B**) Flies were analyzed and treated as in panel A, but they were injured using the standard TBI protocol prior to exposure. Note the higher degree of dispersion in the high ∆ group vs. the low ∆ group (SD = 8.8 vs. 3.1) without injury, which was also present after TBI and exposure to O2 and ISO/O2 in the high ∆ group (SD = 9.0 and 7.7, respectively, vs. 1.9 and 3.6, respectively, for low ∆ group).

**Figure 7 ijms-21-06926-f007:**
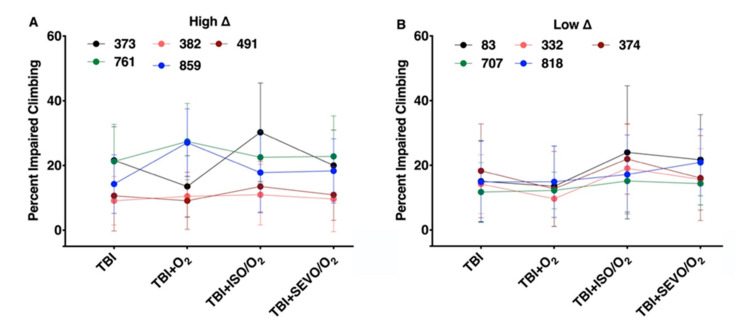
Locomotion response of individual of RAL lines. The effect of treatments on climbing performance were more heterogenous in the (**A**) high ∆ group than the (**B**) low ∆ group.

**Figure 8 ijms-21-06926-f008:**
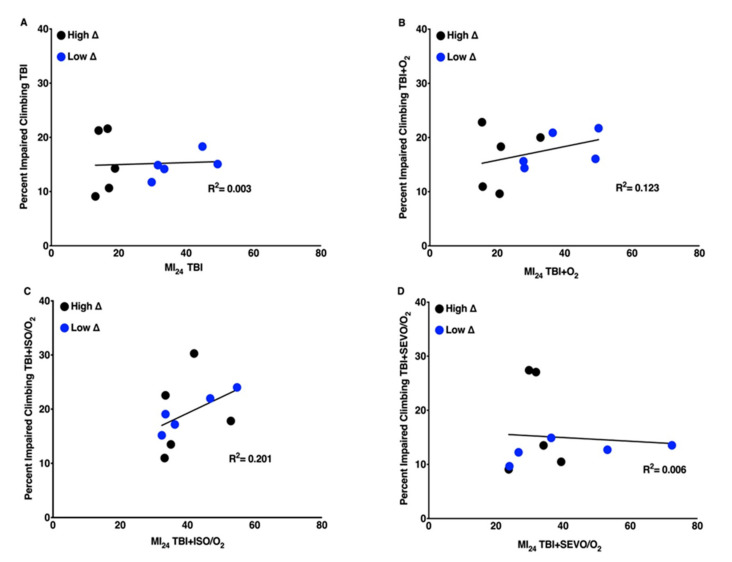
Percent impaired climbing and the MI_24_ are not correlated following exposure to O_2_, ISO/O_2_, or SEVO/O_2_ after TBI**.** Percent impaired climbing as a function of the MI_24_ for (**A**) TBI, (**B**) TBI/O_2_, (**C**) TBI+ISO/O_2_, and (**D**) TBI+SEVO/O_2_. RAL lines in the high ∆ group are indicated by black dots and RAL lines in the low ∆ group are indicated by blue dots.

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
