# Peer review of "Interactions among Genetic Background, Anesthetic Agent, and Oxygen Concentration Shape Blunt Traumatic Brain Injury Outcomes in Drosophila melanogaster"

_ijms, 2020, doi:10.3390/ijms21186926_

Round 1

Reviewer 1 Report

The investigators tested whether several Drosophila lines have altered sensitivities to anesthetics/O2 after TBI using the HIT device developed in Dr. Wassarman's laboratory several years ago. The outcomes measured in the flies were locomotion and mortality 24 hours after trauma. They validated earlier findings that O2 hyperoxia (100% O2) increases mortality and that O2/anesthetics sometimes but not always reduces the effects of hyperoxia, depending on the genetic background. 

While no firm conclusions can be made from these studies, they are important because they suggest that standard of care treatment for TBI with anesthetics/O2 can lead to worth outcomes that varies based on the genetic outcomes of the individual. This paper would have been more interesting if firm conclusions could have been made, such as one anesthetic works better than another anesthetic in reversing the mortality of TBI that is increased by hyperoxia, but it is not surprising to me that no firm conclusions can be made in this regard. 

A previous paper by this group described using Drosophila strains to identify GWAS SNPs that cause increased mortality after TBI + hyperoxia/anesthetics. They identified 3 candidate SNPs that seem to have this effect, but were unfortunately not followed up in this study. Reanalyzing the GWAS results with the new exposures in this paper would have strengthened the paper, such as using KO's or over expressing the candidate genes.

The data was mostly presented as bar grafts which is straightforward but rather bland. As there were a few significant conclusions, a summary figure might be helpful.

In conclusion, this is an interesting study on how standard of care after TBI, namely hyperoxia and anesthesia, might be generally deleterious and more deleterious in some genetic backgrounds more than others. This has implications for treatment of TBI in humans. The paper can be improved by more succinctly and clearly presenting the significant results. 

Author Response

We appreciate the comments made by both reviewers and their detailed examination of our manuscript.

Rev 1

A previous paper by this group described using Drosophila strains to identify GWAS SNPs that cause increased mortality after TBI + hyperoxia/anesthetics. They identified 3 candidate SNPs that seem to have this effect, but were unfortunately not followed up in this study.

Response: Thank you. In order to mechanistically link the SNPs to outcome from TBI, detailed molecular biological experiments will be necessary, which are planned and will take a substantial amount of time.

Reanalyzing the GWAS results with the new exposures in this paper would have strengthened the paper, such as using KO's or over expressing the candidate genes.

Response: In order to reanalyze the GWAS, a sample of ten lines is not sufficient. The hypothesis to be tested was the degree to which sensitivity and resilience to excess death (toxicity) from ISO/O2 translates to O2 and SEVO/O2 exposures.

The data was mostly presented as bar grafts which is straightforward but rather bland. As there were a few significant conclusions, a summary figure might be helpful.

Response: We think that the graphs are the most succinct, understandable representation of the data. Thank you for the suggestion, but we cannot see an easily understandable summary figure.

Reviewer 2 Report

The present manuscript by Scharenbrock and colleagues titled ‘Interactions among genetic background, anesthetic agent, and oxygen concentration shape blunt traumatic brain injury outcomes in Drosophila melanogaster’ tries to study if the genetic background affects the response to hyperoxia or anesthetics in traumatic brain injury Drosophila model. The present work is continuation of previous work by the authors where they showed that the genetic background affects the response to isoflurane (ISO/O2) in traumatic brain injury model. In the present manuscript the authors study the response to sevoflurane by monitoring the mortality index and climbing ability of 10 DGRP lines. The DGRP lines were selected based on their sensitivity or resilience to increased mortality after isoflurane (ISO/O2) exposure. Authors report that the effects of isoflurane observed in brain injury model does not predict the effects of sevoflurane or hyperoxia. Also, authors could not observe any correlation between mortality and locomotor impairment. Authors conclude that the genetic background could influence the potential toxic effects of anesthetics and hyperoxia.

The manuscript is well written however, authors should correct for typos and minor grammatical errors. Authors do mention a caveat that the 10 fly lines studied in the article were selected based on the observations from isoflurane treatment. The results could vary if the authors have included in-between responders to isoflurane treatment, particularly in the case of climbing activity where the uninjured flies show considerable variation in high-delta group. Nevertheless, the results shown in the manuscript shows that the effects of isoflurane and sevoflurane (and hyperoxia) could very well vary among different genetic backgrounds.       

I have following concern:

Authors should confirm if the data shown in Fig. A1 and A14 is not the same as data shown in Fig. 1 and 5.  Also I will encourage the authors to use the same settings for error bar thickness and error bar heads.

Author Response

 Rev 2

We appreciate the comments made by both reviewers and their detailed examination of our manuscript.

The manuscript is well written however, authors should correct for typos and minor grammatical errors.

Response: Thank you for pointing out the errors. We went through the paper carefully and corrected the typos and minor grammatical errors.

Authors do mention a caveat that the 10 fly lines studied in the article were selected based on the observations from isoflurane treatment. The results could vary if the authors have included in-between responders to isoflurane treatment, particularly in the case of climbing activity where the uninjured flies show considerable variation in high-delta group. Nevertheless, the results shown in the manuscript shows that the effects of isoflurane and sevoflurane (and hyperoxia) could very well vary among different genetic backgrounds.       

I have following concern:

Authors should confirm if the data shown in Fig. A1 and A14 is not the same as data shown in Fig. 1 and 5.  

Response: Thank you for noticing. Indeed, Figure 5 was a duplicate of A14. We have replaced Fig 5 and the graph now shows the 24 h data as originally intended. Fig A14 shows the analogous data for 48 h. There are changes in the pattern of climbing impairment between 24 and 48 h. Thank you again.

Also I will encourage the authors to use the same settings for error bar thickness and error bar heads.

Response: We corrected Figure 1. Error bar thickness is uniform now.